## Research Article

parenting; child maltreatment; implementation; faith leaders; Global South

**Corresponding author:**
Lisseth Rojas-Flores;
Email: lrojas@fuller.edu

# Supporting parents in the Global South: implementation of a faith-based parent program in 12 countries

Lisseth Rojas-Flores[1] 🔵, Melanie Ngan[1], Joey Fung[1], Carolyn Casada[1], Patrick Robertson[1], Alex Masibo[2], Briseida Cruz[3], Marion Mortel[4], Ryan Kopper[5], Cherry Marcelo[5], Travis Roberts[5] and Holta Trandafili[5]

[1]School of Psychology and Marriage and Family Therapy, Fuller Theological Seminary, Pasadena, CA, USA; [2]World Vision International, Nairobi, Kenya; [3]World Vision International, San Salvador, El Salvador; [4]World Vision Development Foundation (Philippines), Quezon City, Philippines and [5]World Vision US, Federal Way, Washington DC, USA

## Abstract

Parenting programs are effective ways to reduce child maltreatment and promote nurturing parent–child relationships. Yet, the potential of faith-based, positive parent programs, particularly those conducted globally at scale, remains underexplored. We conducted a pre-post and 6-month follow-up, single-group study of a faith- and community-based parenting program, Celebrating Families (CF), in 12 countries in sub-Saharan Africa, Central America and South East Asia. Using a train-the-trainers model, faith leaders delivered group-based parenting workshops over 3–5 days to a nonrandomized sample of 2201 caregivers across 12 countries. Data was collected at three time points. Shifts in caregiver attitudes and beliefs were assessed pre- and post, and harsh parenting behaviors were measured at pre- and 6-months after CF parent program implementation. Acceptability was demonstrated by high attendance and high satisfaction ratings from facilitators and caregivers. Trained faith and community leaders feasibly delivered the CF parent groups and were rated by caregivers to have strong teaching skills. Qualitative analysis of their feedback at 6-month follow-up highlighted barriers to implementation and areas for improvement. Results with those caregivers who completed the program suggest large to medium effect size improvements in caregiver attitudes around harsh discipline and nurturing parenting by country and change in reported use of harsh parenting behaviors at 6 months. Findings suggest that CF is a feasible and acceptable program with promising short-term effects for caregivers of children and adolescents in low- and middle-income countries.

## Impact statement

Parenting interventions are effective at improving child well-being by reducing harsh and punitive parent beliefs and behaviors and increasing nurturing relationships. However, the implementation of preventive interventions in the Majority World faces a number of challenges, including a lack of culturally sensitive programs, resources, trained facilitators and processes to increase reach and impact of programs and attend to child flourishing outcomes. For this study, we explore whether Celebrating Families, a faith-based parenting program, could be feasibly implemented in 12 sub-Saharan Africa, Central America and South East Asia countries, and whether faith leaders could successfully run the parent groups. Caregivers and facilitators reviewed the program positively and found the attention to their faith to be helpful. Preliminary results also show positive changes in harsh parenting attitudes and beliefs of caregivers immediately after parent workshops as well as a reduction in harsh parenting behaviors 6 months after participating in workshops. This study meets a gap in the implementation evidence by underscoring the capacity of faith-driven parent programs to mobilize local non-specialists and faith leaders toward the holistic well-being of children, effectively shifting harsh parenting attitudes and beliefs to reduce child maltreatment. This study also highlights the significant role of culturally and faith-sensitive parenting practices in bridging community gaps and fostering environments that support parents and children's flourishing in low- and middle-income countries.

## Introduction

Parenting programs in the Global South can reduce harsh parenting and promote nurturing parenting, thereby improving child well-being (Gardner et al., 2016; WHO, 2022). While there is substantial evidence for the use of parenting programs to reduce problem behaviors in children

(Backhaus et al., 2023b; Wang and Zhang, 2023), most preventative parenting studies conducted in low- and middle-income countries (LMICs) have generally focused on young children (ages 1–5) (Jeong et al., 2021). Often middle childhood and adolescent populations (Backhaus et al., 2023a; WHO, 2016) and a strategic focus on positive, youth flourishing outcomes are not included in these interventions (Catalano et al., 2019; Lerner et al., 2021). Although there is emerging evidence from strengths-based perspectives on increasing positive child outcomes, there is a clear need for more culturally and strength-based parenting interventions in global settings. Strength-based family intervention programs demonstrate promising results in international settings by incorporating not only parenting-specific training but also cultural and communal factors related to family life and community engagement. Programs like the Strengthening Families Program (Kumpfer and Magalhães, 2018), the Tuko Pamoja Program (Puffer and Ayuku, 2022) among others delivered in religious settings show promising results in various high- and LMICs (Nicol et al., 2022). These programs highlight the increasing awareness that culturally sensitive programs, particularly those that not only respect but also leverage religious commitment, are needed.

As spiritual health is an often neglected but vital aspect of child well-being (CONSORTIUM, 2022), there remains a need to examine the impact of faith-based programs that incorporate parent and child spirituality. However, to our knowledge, the evidence for faith-based parent programs implemented at scale, especially in LMICs, is nonexistent to date. This study fills the gap by examining the feasibility, acceptability and pre-post changes of a faith-based positive parenting program, Celebrating Families (CF: World Vision International, 2014), implemented by trained local leaders in 12 countries from sub-Saharan Africa, Central America and South East Asia. CF is both faith-based and faith-driven. It is faith-based in that it teaches concepts drawn from the Christian faith (including the use of specific Biblical passages relevant to parenting) and leverages the faith of facilitators and participants in order to promote positive outcomes. It is faith-driven in that several – but not all – desired outcomes include increased spiritual health for both parents and children.

Parenting programs in LMICs are effective strategies to reduce child maltreatment and promote non-violent and nurturing parent–child relationships (Knerr et al., 2013; WHO, 2022). Although the evidence is relatively less robust than that found in high income countries, parenting programs in LMICs have been shown to reduce physical and emotional abuse (Wang and Zhang, 2023). Additionally, parenting programs in LMICs have been shown to foster nurturing parent–child relationships by improving parental knowledge and bolstering caregiver social support (Wang and Zhang, 2023; WHO, 2022). Overall, the existing evidence underscores the unique challenges of implementing parenting programs in LMICs and calls for continued evaluation and adaptation of such programs to address complex sociocultural factors.

The perception of safety in relationships is essential for child flourishing and positive development. In LMICs, many families live in contexts of adversity. Children and youth often face structural violence brought on by poverty and exposure to violence in and out of the home (Know Violence in Childhood, 2017); Ward et al, 2015). A vast body of research reveals the detrimental effects of violence on children across physical, social, cognitive and emotional health domains (Akande, 2000; Devries et al., 2017). In contrast, there is evidence that greater family connection is associated with a higher prevalence of flourishing among adolescents in global settings (Whitaker et al., 2022). However, safety in

relationships and environments continues to be neglected in studies of child thriving and flourishing (Ettinger et al., 2022).

Spiritual health is another often overlooked aspect of holistic well-being. Spirituality and faith are recognized as important determinants of positive child development (CONSORTIUM, 2022). They have been shown to protect against stress, build resilience (Kim, 2008; Salas-Wright et al., 2013) and provide meaning and purpose for children growing up in difficult circumstances (Regnerus, 2003; Tirrell et al., 2019). Studies also found that spirituality promotes life satisfaction (Holder et al., 2016; Tirrell et al., 2019) and prosocial traits such as kindness and empathy among children in global contexts (Leman et al., 2017).

There is a growing global interest in parenting interventions that are culturally and religiously-sensitive. As over eight out of every 10 adults and children globally endorse a faith or religious affiliation (Pew Research Center, 2012), faith and religiosity are vital to include when implementing parent programs internationally. An estimated 62% report a Christian religion affiliation in sub-Saharan Africa (Pew Research Center, 2012) and 97.5% in Central America (Johnson and Grim, 2020). The Philippines is the third-largest Catholic population in the world (Lipka, 2015). Caregivers can influence positive child development through the transmission of healthy spiritual and religious beliefs and practices (Baring et al., 2016). Parenting programs that incorporate the spiritual nurturance of children may bolster the effectiveness of interventions targeting healthy child development.

Without doubt, religion can be used to justify harmful parenting practices. However, religious beliefs have the potential to transform entrenched parenting beliefs, attitudes and even practices (Petro et al., 2017). For example, within the Judeo-Christian faith traditions, teachings that are particularly conducive to this process include the Christian concept of the Imago Dei teaches that all people, including parents and children, are made 'in the image of God', affirming their inherent dignity and worth. Within Christian moral teachings, parents are seen as primarily responsible for all aspects of their children's well-being. These teachings also position children as gifts from God and as an integral part of the Christian community. This motivation can encourage parents to adopt behaviors promoting child well-being and flourishing, such as by creating safe home environments (El-Khani and Calam, 2018).

A key way to transform the lives of children and their families is by implementing parenting programs in real-world settings. While implementing parenting programs in LMICs contributes to a larger global agenda (e.g., UN-SDGs, 2024), scaling up such programs in these contexts is often challenging due to lack of professionals available to implement them. However, evidence shows that trained non-professionals can produce almost the same benefits as professional-led groups, and train-the-trainers models show promise for bridging this gap (Tomlinson et al., 2017). The involvement of faith and other community leaders who possess convening power and are vital agents of community change can leverage their influence to change attitudes on corporal punishment and healthy parent–child relationships in highly effective ways that transcend cultural norms while remaining sensitive to the local context (Robinson, 2010; Rutledge and Eyber, 2019). Moreover, taking advantage of the existing service delivery infrastructure of large international development and non-governmental organizations (INGOs) can assist in capacity building and scaling interventions in LMICs.

In partnership with a large Christian international development INGO with its established service delivery infrastructure, World Vision International (WVI) and informed by a socioecological

model (Bronfenbrenner, 1979), this study seeks to examine the process of implementation and scaling of a faith-based parenting program in LMICs. CF is a faith-based parenting program developed by WVI (World Vision International, 2014) designed to reduce child maltreatment, enhance positive and spiritual parent child activities and promote gender equity in the home.

Using archival data collected from June 2022 to March 2023, this study examines the feasibility, acceptability and pre-post changes associated with the implementation of the CF faith-based parenting program in 12 countries in the Global South. We aimed to assess (1) the acceptability of the parent program for caregivers and facilitators, through caregiver attendance, attrition and satisfaction with the program, we hypothesized that caregivers would find the program acceptable; (2) the implementation process (feasibility) and the extent to which facilitator training and parent workshops were implemented as described. This was based on facilitator feedback on their perceived knowledge improvement following training and perceived barriers and recommendations for improvement/sustainability and (3) the immediate pre-post changes in positive parenting and harsh discipline attitudes and 6-month changes in harsh parenting behaviors. We hypothesized that caregivers would rate improved scores on positive parenting beliefs and reduced endorsements of harsh parenting beliefs and behaviors across time.

## Methods

### Study setting

Parent groups were implemented in 12 countries in Africa, Central America and Southeast Asia. These included seven countries in sub-Saharan Africa (the Democratic Republic of the Congo [DRC], Ethiopia, Ghana, Kenya, Mozambique, Rwanda, and Zimbabwe), one in South East Asia (Philippines) and four in Central America (El Salvador, Guatemala, Honduras and Nicaragua). Although there is significant social, political and cultural diversity across the countries, the circumstances for youth in these regions shared some similarities. Ten countries fall in the medium youth development categories according to the Global Youth Development Index (Commonwealth Secretariat, 2021), while DRC and Mozambique fall in the low youth development category. Moreover, all 12 countries fall within levels 2 and 3 of Rosling's four levels of income (Gapminder, 2023; Rosling et al., 2018). These countries share similar challenging developmental environments for children regarding education, employment opportunity, health and well-being, equity and inclusion, political and civic participation and peace and security.

### Selection process

This study employed a purposive, non-randomized, non-probability sample. There was no blinding of participants employed.

*Caregivers*: Eligible parents and caregivers were recruited from families that had participated in WV activities in each country (10–25 parents per group). Inclusion criteria for parent/caregiver were (1) age 18 or older and (2) primary caregiver of a child in the household aged 7–18. Targeted child respondents were recruited from all participating families. If there was more than one eligible child in the household, the participating adult was asked to identify one child only. Inclusion criteria for child respondents were (1) aged 7–18 years, (2) one caregiver participating in the study and (3) lived in the house with participating caregiver. Adults and

children were excluded if they exhibited acute mental health problems or if the caregiver had participated in another parenting intervention in the past 12 months. Families received no compensation. Adult provided consent and children provided verbal assent. Throughout this paper, 'parent' includes adult caregivers even if they are not the biological parents, and 'target child' refers to the participating child/adolescent.

Participant demographic characteristics are presented in Table 1. A total of 2201 parents and 2035 target children participated in the study; 1741 parents completed the post-measure. The mean age of the target child was 10 years, 8 months (range = 7–18; SD = 2.7), and the majority were female (63%). Over one-third (35.8%) of participating families were considered economically disadvantaged per Multidimensional Poverty Index (MDPI). For caregiver attrition by region, see Supplementary Figures S1–S3 in the Supplemental Materials.

*Facilitators*: A total of 210 volunteer facilitators were identified by local leaders and WV staff and recruited from local churches and organizations to lead parent workshops. With its established service delivery infrastructure, the Christian International Development INGO had existing partnerships with local communities and faith actors with whom they have previously trained on the CF parenting program or other technical workshops (e.g., microfinance and livelihood). While the participating countries operate in unique contexts, the identification and selection of facilitators was focused mainly on faith leaders with the ability to communicate in English and the local language and who demonstrated strong group facilitation skills. All facilitators were trained or given a refresher training on the CF curriculum, ranging from 1 to 3 days. The mean age of facilitators was 43.4 years (range = 19–70). Gender parity differed by region. Central America had a majority of female facilitators (88.6%), while Africa had a majority of male facilitators (62.5%). In terms of their role in the community, across all three regions 64% ($n = 65$) of the facilitators identified themselves as Faith Leaders while 36% ($n = 37$) were Community Leaders. Specifically, in Africa 69% ($n = 45$) facilitators were Faith Leaders while 31% ($n = 20$) were Community Leaders. All of the facilitators ($n = 12$) in the Philippines were Faith Leaders; while in Central America, 32% ($n = 8$) of the facilitators were Faith Leaders and 68% (17) were Community Leaders (see additional facilitator demographic information in Table 1).

### Data collection processes

A mixed-methods, pre-post and 6-month follow-up design was utilized in this study. Data was collected at three time points: (1) baseline/pre-workshop (T1, around July 2022), (2) immediately following the parenting workshop (T2) and (3) 6 months after the parent program (T3, around March 2023). Following the conclusion of the CF parent workshop and T2 survey, caregivers continued to meet at least once per month, in peer support groups; these groups were implemented for 6 months. This paper focuses only on caregivers who completed parent program and have data available at T1, T2 and T3 and group facilitator T3 data.

To account for varying levels of literacy, trained local data collectors (enumerators) verbally administered consent and assent forms and all questionnaires. Enumerators entered participants' responses on smartphones or laptops to Kobo Toolbox (kobotoolbox.com, n.d.) an open-source mobile data collection software.

**Table 1.** Participant demographic characteristics

| Variable | Total Fac. (n = 161) | Total Car. (n = 2201) | Total Child (n = 2035) | Africa (7 countries) Fac. (n = 104) | Africa (7 countries) Car. (n = 1277) | Africa (7 countries) Child (n = 1240) | Central America (4 countries) Fac. (n = 44) | Central America (4 countries) Car. (n = 725) | Central America (4 countries) Child (n = 640) | Asia (1 country) Fac. (n = 13) | Asia (1 country) Car. (n = 199) | Asia (1 country) Child (n = 155) |
|---|---|---|---|---|---|---|---|---|---|---|---|---|
| Age | 43.4 (11.1) | 40.6 (14.0) | 10.8 (2.7) | 44.4 (9.2) | 42.4 (13.5) | 10.8 (2.8) | 35.4 (9.6) | 37.5 (14.1) | 10.6 (2.5) | 57.4 (11.2) | 39.1 (14.7) | 11.8 (2.6) |
| Sex | | | | | | | | | | | | |
| Female | 87 (54.0%) | 1512 (68.7%) | 1145 (56.3%) | 39 (37.5%) | 768 (60.1%) | 684 (55.2%) | 39 (88.6%) | 607 (83.7%) | 372 (58.1%) | 9 (69.2%) | 137 (68.8%) | 89 (57.4%) |
| Religion | | | | | | | | | | | | |
| Christian | 155 (96.3%) | 1802 (81.9%) | 1849 (90.9%) | 102 (98.1%) | 1045 (81.8%) | 1131 (91.2%) | 42 (95.5%) | 584 (80.6%) | 584 (91.3%) | 11 (84.6%) | 173 (86.9%) | 134 (86.5%) |
| Muslim | 0 (0.0%) | 97 (4.4%) | 90 (4.4%) | 0 (0.0%) | 96 (7.5%) | 90 (7.3%) | 0 (0.0%) | 1 (0.1%) | 0 (0.0%) | 0 (0.0%) | 0 (0.0%) | 0 (0.0%) |
| Buddhist | 0 (0.0%) | 1 (0.1%) | 0 (0.0%) | 0 (0.0%) | 1 (0.1%) | 0 (0.0%) | 0 (0.0%) | 0 (0.0%) | 0 (0.0%) | 0 (0.0%) | 0 (0.0%) | 0 (0.0%) |
| Hindu | 0 (0.0%) | 1 (0.1%) | 0 (0.0%) | 0 (0.0%) | 0 (0.0%) | 0 (0.0%) | 0 (0.0%) | 0 (0.0%) | 0 (0.0%) | 0 (0.0%) | 1 (0.5%) | 0 (0.0%) |
| Non-religious | 1 (0.6%) | 118 (5.4%) | 33 (1.6%) | 0 (0.0%) | 77 (6.0%) | 11 (0.9%) | 1 (2.3%) | 39 (5.4%) | 22 (3.4%) | 0 (0.0%) | 2 (1.0%) | 0 (0.0%) |
| Other | 5 (3.1%) | 40 (1.8%) | 63 (3.1%) | 2 (1.9%) | 7 (0.5%) | 8 (0.6%) | 1 (2.3%) | 23 (3.2%) | 34 (5.3%) | 2 (15.4%) | 10 (5.0%) | 21 (13.5%) |
| Household | Characteristics | | | | | | | | | | | |
| MDPI | 0.27 (0.21) | | | 0.29 (0.2) | | | 0.27 (0.25) | | | 0.27 (0.20) | | |
| MPI-Poor | 35.8% | | | 40.0% | | | 38.9% | | | 34.9% | | |
| Location | | | | | | | | | | | | |
| Urban | 313 (12.5%) | | | 241 (18.8%) | | | 18 (2.6%) | | | 54 (29.0%) | | |
| Peri-Urban | 379 (15.2%) | | | 46 (3.6%) | | | 226 (32.7%) | | | 33 (17.7%) | | |
| Rural | 1803 (72.3%) | | | 993 (77.6%) | | | 448 (64.7%) | | | 99 (53.2%) | | |

*Notes*: Fac.: facilitators; Car.: caregivers; MDPI: Multidimensional Poverty Index; MPI-Poor: percentage of households considered MPI poor.

## Intervention: CF curriculum

The CF parenting program is a 3–5-day group-based manualized curriculum that integrates best practices for family-based parenting with Christian faith principles. The program seeks to reduce child maltreatment and the use of harsh parenting practices such as corporal punishment, while increasing parent–child positive relationships and child-flourishing outcomes. It utilizes a strengths-based approach and seeks to ensure that all families have hope for the future, recognize harm from their pasts, are empowered with agency, and experience loving and gender-equitable spousal and caregiver–child relationships (see Table 2). The core principles underlying the program are theoretically informed by attachment theory (Thompson, 2024), intergenerational trauma (Starrs and Békés, 2024) and family systems theory (Becvar et al., 2023). The Christian faith principles of the CF curriculum include a focus on forgiveness, grace and reconciliation (as modeled in the biblical Parable of the Prodigal Son, referenced in the training); the theological concept of the Imago Dei; teaching that all people are created in the image of God and thus accorded equal dignity and worth; and servant-hearted leadership that utilizes discipline to guide and protect, rather than punish, as taught in the Book of Proverbs and the New Testament (see Table 2; CF Parent Program Key Components).

The curriculum was developed in 2011 and revised with local input from various countries (World Vision International, 2014). There is strong qualitative evidence regarding the benefits of the CF curriculum and methodology in various global settings. Most recently, qualitative case studies from Afghanistan, Myanmar and Tanzania with different family structures (e.g., nuclear, multigenerational) showed that CF has positive effects at the family level and challenges harmful cultural and social norms (Barett and Niyonkuru, 2019). The curriculum has been adapted for many cultural contexts and languages, including Swahili, Amharic, Tagalog, Arabic, Spanish and French.

WV utilizes a train-the-trainer model to train local facilitators (e.g., faith or community leaders) in the manualized curriculum. Selected WV staff, local partners, faith leaders and community influencers participated in the training of facilitator (ToF) workshops led by certified CF trainers face-to-face over 5 days. During the training, facilitators were exposed to the full CF curriculum and learned how to deliver the workshops in the community. All facilitators received a certificate of completion and answered a survey about their level of satisfaction with the training.

The CF parent workshop was delivered in groups by these unpaid trained facilitators. Groups of 15–25 parents met in community venues, such as churches or community centers. The program included follow-up activities, such as peer support groups with participating parents to support behavioral change. In the present study, these groups were implemented for 6 months following the conclusion of the CF parent workshop.

This project was conducted as part of ongoing program evaluation efforts. Internationally recognized ethical guidelines for research with children (Graham et al., 2013) were followed, including obtaining parental consent and child assent before data collection, maintaining confidentiality and ensuring that participants (including children) had the right to withdraw their participation at any point. Ethical approval for archival data analyses was obtained from the Fuller Theological Seminary Human Subjects Review Committee (IRB 19333854-1).

**Table 2.** Celebrating families parent program key components

| "Celebrating families" Title of session | Parenting practices/ principles | Activities |
| --- | --- | --- |
| 1. Welcome/Setting Expectations | | Introduction to CF framework |
| 2. Hopes and Dreams/Aspiration for Family | Identify aspirations and hope for family; family values | Draw hopes and dreams for family |
| 3. Childhood Experiences Influence on Parenting | Parent reflection on past parenting experiences | Draw childhood experiences |
| 4. Wholeness vs. Brokenness/Positive & Negative Experiences | Assessment of family of origin and current family experiences | Identify group positive and negative experiences |
| 5. Inherent Goodness in Self and Others | Positive attention (quality time, praise, words of affirmation, etc.) | Discuss how to show love |
| 6. Positive and Harsh Parenting | Perspective taking from the vantage point of children and parents | Discuss how God equally values children and parents |
| 7. Grace/Positive Discipline Strategies | Positive and non-violent discipline strategies | Discuss positive discipline in the light of faith and research |
| 8. Thanksgiving and Forgiveness for Parents | Self-compassion, conflict resolution and gratitude and forgiveness | Reflect on strengths/areas of growth as a parent |
| 9. Family Defining Moments | Realistic expectations and hopes | Create family timeline |
| 10. Building Families on Firm Foundation | Problem-solving and planning | Writing practical steps for caregivers |
| 11. Developmentally Sensitive Parenting | Self-assessment of current parenting behaviors, awareness of developmental stages | Identify start, continue and stop behaviors |
| 12. Committing to Nurturing Parenting | Nurturing behaviors, affirmations from parent to child | Create tangible affirmations for children |
| 13. Blessing Parents, Families and Communities | Individual and communal affirmations and social support | Express affirmations for each group member |
| 14. Applying What We've Learned | Problem-solving and goal-setting | Set SMART goals |

## Measures and instruments

Questionnaires were translated into the respective majority language of each country, including French, Spanish, Tagalog and Portuguese, by a group of qualified WV staff and hired translators. All translators were competent in both languages in each country. Translations were revised by regional project coordinators and the researchers. Questionnaires were pre-tested with a small sample of caregivers in each country.

### Demographic variables

At baseline, caregivers provided demographic information, including age, gender, religion, poverty level, and household location. Poverty was measured using the Global Multidimensional Poverty Index (MDPI) (Oxford Poverty & Human Development Initiative, 2018), which measures the incidence and intensity of poverty over three dimensions (health, education and living standards). Household location indicated whether the family resided in an urban, rural, or peri-urban area.

### Implementation fidelity measures

Implementation fidelity of the ToF and CF workshops was measured through daily checklists. Trainers and facilitators reported whether they adhered to implementation procedures, such as following selection criteria, completing an attendance list, and/or providing childcare and food. Also, when an aspect was not implemented, facilitators were asked to report why (see Supplementary Table S7 – *Facilitator Self-Reported Implementation Fidelity for CF Parent Workshops*, found in Supplementary Materials). Parent attendance for each workshop was collected using attendance sheets completed by facilitators. Scans of these sheets were uploaded to Kobo Toolbox, and data was entered and analyzed by the research team.

### Acceptability measures

At the T2 post-workshop, parents reported their satisfaction, using a 5-item Likert scale to rate the helpfulness of: the content of the program, group leaders' teaching and leadership skills, group discussion and interaction and attention given to their religious beliefs. Parents were also asked if they would recommend the workshop to a friend using a yes/no response set. Finally, they responded to two open-ended items: (1) "What was most helpful about the program?" and (2) "What was the least helpful about the program?" Similarly, facilitators provided feedback on program implementation barriers and challenges via an online anonymous survey. Open-ended responses were coded to identify main themes (see Table 4).

*Positive and harsh parenting attitudes and beliefs.* A Celebrating Families Workshop Survey (World Vision International, 2020), consisting of 17-items assessing parent knowledge, beliefs and attitudes about parenting was administered at baseline (T1) and immediately following the parent workshop (T2). Exploratory and Confirmatory Factor Analyses were conducted to ensure the factor structure was invariant across regions and countries. CFAs were conducted separately at each time point. Models yielded a good fit for a two-factor model (CFI = .988; TLI = .984; RMSEA = .045, 95% CI = [.040, .050]): Harsh Parenting Attitudes and Beliefs (HshP) and Positive Parenting and Spiritual Nurture Attitudes and Beliefs (PsP). Items were rated on a 5-point Likert scale (1 = *strongly disagree*, 5 = *strongly agree*).The 5-item HshP scale assessed attitudes toward corporal punishment and use of violent parenting practices (e.g., shouting at the child, hitting; T1 $\alpha$ = .76, T2 $\alpha$ = .73). The 7-item PsP scale included items such as spending time with children and family and the importance of spiritual nurturance and virtues such as gratitude (T1 $\alpha$ = .77, T2 $\alpha$ = .78).

*Harsh parenting behaviors.* Using the *MICS – Child Discipline Module for children age 5–17* (UNICEF, 2019), the most widely used assessment of child disciplinary practices in LMICs (Akmatov, 2011), caregivers reported their use of violent discipline and harsh parenting practices (eight items, including shouting, slapping, called demeaning names, hitting with an object, beating) at time 1 and time 3 (6 months after parent program). These practices were summed to create a continuous score.

### Data analyses

### Qualitative analyses

Two open-ended responses to consumer satisfaction items were translated from the various languages into English using ChatGPT 3 and verified by bilingual research assistants. A considerable amount of repeated themes were noted by coders. Therefore, given the large caregiver sample size and to ensure quality, manageability and feasibility of qualitative analyses, we elected to code a random, representative subset of caregiver responses (33%; see Table 4). The caregiver response subsets were randomly selected from each region to ensure the subset was representative and key themes from each region could be identified. All facilitator responses were coded (see Table 3). Responses were analyzed in two stages based on grounded theory (Strauss and Corbin, 1998) using Dedoose, a web-based qualitative research tool (Dedoose, 2023). Three coders read transcripts and categorized responses into major themes and subthemes. Four other independent coders then reviewed and coded participant transcripts using the identified themes/subthemes. Coders were of diverse ethnic-racial backgrounds. Discrepancies were resolved through consensus. Interrater reliability for caregiver satisfaction ($\kappa$ = 0.78) and facilitators' perceived barriers ($\kappa$ = 0.93) indicated substantial agreement (Landis and Koch, 1977). Frequency (F; total number of times a theme was mentioned) and extensiveness (E; total number of participants who commented at least once about a theme) were calculated by region (see Tables 3 and 4).

### Quantitative analyses

To examine program acceptability, descriptive statistics were calculated. Paired *t*-tests comparing pretest to posttest results were stratified and implemented by region and then by country of participation (see Table 5). For each variable of comparison, listwise deletion was applied. To control for the inflated Type-I error rate potentially due to the multiple comparisons of the two key variables, we set the planned alpha level at .01 instead of .05. Cohen's *d* was calculated as standardized effect size measures. All analyses were conducted in R and SPSS.

### Results

### Attendance

Of the 2199 caregivers who completed baseline assessments, 495 did not participate in the CF parenting program and were, therefore, excluded from the study analyses. Caregiver attendance at parent group workshops was closely monitored. Across all three regions, 24.69% of participants attended less than 50% of CF sessions, 6.28% attended 50-74% of sessions, 2.96% attended 75-99% of sessions, and 66.08% attended all sessions. Overall, 69.04% of caregivers demonstrated high attendance (over 75% of sessions); these attendance rates are only limited to those who started the parent program. For a more detailed attendance breakdown by each region, see Supplementary Table S6 and Supplementary Figures S1–S3 in the Supplemental Materials. Attendance was

**Table 3.** Parent satisfaction frequency and extensiveness for 'Most Helpful' responses

| Theme | Total | | Africa | | Asia/Philippines | | Central America | |
|---|---|---|---|---|---|---|---|---|
| | Freq (%) | Ext (%) | Freq (%) | Ext (%) | Freq (%) | Ext (%) | Freq (%) | Ext (%) |
| Positive parenting | 37.5 | 34.2 | 38.1 | 30.1 | 33.9 | 35.7 | 37.5 | 42.7 |
| Positive attitudes and knowledge | 17.1 | 14.3 | 20.0 | 15.1 | 11.5 | 14.0 | 13.5 | 12.9 |
| Positive practices | 8.3 | 7.2 | 10.1 | 7.2 | 12.7 | 16.3 | 3.4 | 3.9 |
| Faith-based parenting | 12.1 | 12.1 | 8.0 | 7.9 | 9.7 | 5.4 | 20.6 | 26.0 |
| Positive family relations | 32.5 | 30.3 | 39.2 | 33.4 | 40.6 | 42.6 | 17.1 | 19.0 |
| Positive community relationships | 3.0 | 2.8 | 3.4 | 2.9 | 1.8 | 2.3 | 2.7 | 2.8 |
| Parent personal growth | 3.1 | 2.5 | 2.0 | 1.7 | 2.4 | 2.3 | 5.3 | 4.5 |
| Parent program quality | 19.0 | 26.2 | 10.0 | 26.7 | 19.4 | 14.7 | 35.8 | 29.1 |
| Program content | 13.4 | 10.3 | 16.3 | 11.8 | 7.9 | 6.2 | 9.9 | 8.4 |
| Group dynamics | 0.5 | 0.4 | 0.6 | 0.7 | 0.0 | 0.0 | 0.2 | 0.3 |
| Overall positive comment | 13.4 | 9.3 | 9.6 | 6.5 | 6.7 | 4.7 | 23.0 | 17.3 |
| No feedback | 5.0 | 4.0 | 7.4 | 5.2 | 1.8 | 2.3 | 1.7 | 2.0 |

*Notes:* Freq: standardized frequency = number of times a code was used in a particular region/total number of codes used in that region; Ext: standardized extensiveness = number of participants who used a certain code at least once/total number codes used at least once.

**Table 4.** Barriers reported by group facilitators – frequency and extensiveness percentages

| Theme | Total | | Africa | | Asia/Philippines | | Central America | | Quotes |
|---|---|---|---|---|---|---|---|---|---|
| | Freq | Ext | Freq | Ext | Freq | Ext | Freq | Ext | |
| Logistical challenges | 38.9 | 33.5 | 43.6 | 36.5 | 31.1 | 22.1 | 26.0 | 24.9 | 'Transportation is needed because the groups are not in the neighborhood'. Rwanda, Female, 31 |
| Parental participation | 13.8 | 13.6 | 12.6 | 12.6 | 15.2 | 12.5 | 17.0 | 17.1 | 'The challenge was to bring the families together due to their work commitments; the men did not attend'. Nicaragua, Female, 31 |
| Training needs | 9.2 | 14.7 | 7.2 | 12.1 | 17.4 | 25.0 | 13.5 | 19.5 | '(Include) different training like economic development training'. Ethiopia, Male, 39 |
| Parental factors | 5.7 | 6.0 | 4.9 | 5.3 | 7.4 | 9.6 | 7.6 | 7.2 | 'The other challenge is about parents who could neither read nor write. They could simply participate orally or by drawing, if the exercise allowed'. Mozambique, Male, 32 |
| Financial barriers | 7.1 | 6.8 | 9.1 | 8.6 | 4.8 | 4.8 | 1.3 | 1.7 | 'Incentives during training to ensure that family concerns such as food do not hinder their time commitment'. Philippines, Male, 65 |
| Cultural challenges | 2.9 | 3.6 | 3.4 | 4.4 | 1.5 | 2.9 | 1.5 | 1.4 | 'Cultural belief that women should not stand in front of men'. Kenya, Female, 43 |
| Community engagement | 1.9 | 2.2 | 2.0 | 3.0 | 0.4 | 1.0 | 2.0 | 2.7 | 'I think the LCC model should include chiefs and opinion leaders in the community'. Ghana, Female, 23 |
| Facilitator Limitations | 1.8 | 3.5 | 1.5 | 3.0 | 2.6 | 4.8 | 2.6 | 4.8 | 'I need to receive effective training in stress management'. DRC, Male, 31 |
| No feedback | 18.8 | 16.1 | 15.6 | 14.7 | 19.6 | 17.3 | 28.5 | 20.8 | 'No challenges arose during the training'. Zimbabwe, Male, 50 |

*Notes:* Freq: standardized frequency = number of times a code was used in a particular region/total number of codes used in that region; Ext: standardized extensiveness = number of participants who used a certain code at least once/total number codes used at least once.

not available for Kenya. Reasons for absence included parental ill health, work, and childcare commitments.

### Acceptability and fidelity

*Facilitator satisfaction with training*: Facilitator's satisfaction with their training and overall experience were rated on a 5-point Likert scale, with higher scores representing greater satisfaction. Facilitators ($N = 161$) across the three regions rated their overall training experience highly (Africa: $M = 4.52$, $SD = 0.59$; Central America: $M = 4.59$, $SD = 0.58$; the Philippines: $M = 4.09$, $SD = 0.79$). Facilitators rated the clarity and sufficiency of instructions and materials similarly. Additionally, facilitators reported perceived improvement in knowledge regarding the key messages of the curriculum, including positive parenting, non-violent discipline and child-safeguarding strategies and the overall spiritual nurture of children. Overall, the facilitator satisfaction with the program is captured by a male faith leader facilitating groups in Ethiopia: '*This project is useful for our community, children, and parents*'.

*Caregiver satisfaction with parent program*: Across all regions, parents reported high levels of satisfaction with the overall content of the program (Africa: $M = 4.79$, $SD = 0.43$; Central America: $M = 4.43$, $SD = 0.52$; the Philippines: $M = 4.50$, $SD = 0.51$). When asked if they would recommend the program to a friend, 54.3% of caregivers across all regions responded 'definitely yes', and 43.8% responded 'yes'. Additionally, a majority of caregivers rated facilitators' teaching skills (Africa: 98.71%; Central America: 99.09%; the Philippines: 96.75%) and leadership skills (Africa: 98.39%; Central America: 95.00%; the Philippines: 96.75%) as helpful or extremely helpful. The remaining 1.9% responded 'maybe', 'no', 'definitely no' or were missing responses. With regards to appropriateness, parents found the attention to their religious beliefs to be very helpful (Africa: $M = 4.62$, $SD = 0.57$; Central America: $M = 4.34$, $SD = 0.53$; the Philippines: $M = 4.53$, $SD = 0.51$).

Data from checklist forms were processed by country and region. Overall, the implementation of both the ToF and CF workshops followed the required protocols. Four countries noted delays in the printing process of certificates for facilitators, consistent with the complaints regarding the lack of certificates in the facilitator feedback.

### Pre-post caregiver change

*Parenting attitudes and knowledge pre- to post-parent program (T2 − T1)*: The final analyses included 2021 caregivers who attended all CF sessions and completed the pre- and post-evaluation scale. To analyze change, we compared scores of positive and harsh parenting attitudes and knowledge before and after receiving the parent program (see Table 5). For caregivers in all countries in Africa, paired *t*-tests on the HshP scale showed on average a decline of −3.71 points (range = 0–20; $Mpre = 7.04$; $Mpost = 3.33$), corresponding to a very large effect size (Cohen's $d = 1.00$; Sawilowsky, 2009). Conversely, African caregivers on the Positive/Nurturing Parenting scale (PsP) scale showed a medium to large effect size (Cohen's $d = 0.70$) as evidenced by an increase of 3.68 points (range = 0–28; $Mpre = 17.63$; $Mpost = 21.3$). This value corresponds to a Cohen's $d$ of 0.70, which are considered a 'medium to large' effect size. In Central America, the paired *t*-tests for the Harsh Parenting showed on average observed a significant decline of −1.25 points (range = 0–20; $Mpre = 4.40$; $Mpost = 3.15$; $p < 0.001$, $|d| = 0.41$), while the on the PsP Parenting scale on average observed an increase of 3.49 points (range = 0–28; $Mpre = 18.02$; $Mpost = 21.50$; $p < 0.001$, $|d| = 0.62$). Cohen's $d$ of 0.41 and 0.62 were 'medium' effect sizes. Lastly, the paired *t*-tests, for caregivers in the Philippines showed on average observed, a significant decline of −2.26 points (range = 0–20; $Mpre = 8.06$; $Mpost = 5.80$); $p < 0.001$, $|d| = 0.59$) for the Harsh Parenting scale; whereas on the PsP scale on average observed an increase of 1.58 points (range = 0–28; $Mpre = 15.99$; $Mpost = 17.57$; $p < 0.001$, $|d| = 0.32$). Cohen's $d$ of 0.59–0.32 are considered a 'medium to small' effect size.

*Harsh parenting behaviors pre- to 6 months (t3) after parent program (T3 − T1)*: We also compared scores of harsh parenting behaviors endorsed at baseline and 6 months (t3) after caregiver participation in CF parent program. In all countries in Africa, paired *t*-tests on the harsh parenting behaviors scale, showed an average decline of 1.26 points (range = 1–8; $Mpre = 1.84$; $Mt3 = 0.58$), corresponding to a medium to large effect size (Cohen's $d = 0.73$; Sawilowsky, 2009). For the caregivers in the Philippines, paired *t*-tests, showed on average observed, a significant decline of 0.97 points (range = 1–8; $Mpre = 2.16$.; $Mt3t = 1.18$);

$p < 0.001$, $|d| = 0.65$). Lastly, in Central America, the paired *t*-tests for the Harsh Parenting Behaviors showed on average observed a significant decline of 0.99 points (range = 1–8; $Mpre = 1.10$; $Mt3 = 0.11$; $p < 0.001$, $|d| = 0.94$) (see harsh parenting behavior mean differences by country in Table S8 in the Supplementary Materials).

In addition, we included other secondary outcome measures at baseline and 6-month follow-up, including measurement of caregiver psychological distress and social support, as well as child report of parent use of harsh parenting behaviors and child self-assessment of flourishing outcomes such as hope for future, positive relationship with family and peers among others. Preliminary analyses indicate outcomes are in the hypothesized direction (Rojas-Flores et al, n.d.).

### Facilitators' perceived barriers for program implementation

Group facilitators suggested lessons for program implementation (see Table 4). Across all three regions, the top main barriers reported by facilitators included (1) logistical challenges, ranging from transportation challenges to bad weather to lack of materials (F = 38.9%, E = 33.5%); (2) challenges regarding parental participation in the program due to work schedules, repeated tardiness and a desire for involvement of both parents (F = 13.8%, E = 13.6%); and (3) needs for training beyond parenting, including economic training, training for certain developmental stages (i.e., adolescence) and training in navigating interpersonal conflict (F = 9.2%, E = 14.7%). One facilitator's response summed up barriers encountered during this pilot study well: 'It's a new experience for our community to come together for talks, and the changes are not easy, but we did our best' (Honduras, Male).

Notably, as an international development agency, WV provides other activities in communities within which it collaborates. Of the eleven possible WV activities provided in communities participating in this study, five were most endorsed: child sponsorship, education (which may be for adults or children), child protection, child participation (in other activities such as clubs) and spiritual nurture programs (see Table S9 in Supplemental Materials).

### Discussion

This single-group study is among the first to examine the implementation and pre-post and 6-month follow-up changes in parent outcomes of a faith-based parenting program in twelve sub-Saharan African, South East Asian and Central American countries. Overall, with optimistic caution, the results suggest that the CF program is highly feasible to implement by trained faith leaders and well accepted by parents. We found medium to large immediate pre-post changes in positive parenting and harsh parenting attitudes and beliefs as well as harsh parent behaviors at 6 months after the CF parent program. These findings excluded participants that did not received the CF parent program and could suggest larger effects than would be expected in a pre-post design. Notwithstanding these limitations, results provide a strong rationale for future rigorous studies to examine the efficacy and causality of this novel faith-based parent program.

*Acceptability and appropriateness*: We explored the acceptability of this program and related factors of sustainability by soliciting parents' feedback of the program. Overall, caregivers across the three regions highlighted that the emphasis on positive parenting and family relationships were most helpful (see Table 3).

**Table 5.** Harsh and positive parenting attitudes and knowledge outcomes

| Harsh parenting | n | Pre-test M (SE) | Post-test M (SE) | M diff. | p | Cohen's d |
|---|---|---|---|---|---|---|
| Africa | 1177 | 7.04 (0.13) | 3.33 (0.09) | −3.71 | <0.001 | 1.00 |
| DRC | 163 | 7.66 (0.28) | 3.53 (0.22) | −4.12 | <0.001 | 1.30 |
| Ethiopia | 153 | 8.07 (0.39) | 2.88 (0.21) | −5.18 | <0.001 | 1.31 |
| Ghana | 192 | 7.81 (0.33) | 2.93 (0.21) | −4.88 | <0.001 | 1.26 |
| Kenya | 195 | 8.06 (0.30) | 3.79 (0.25) | −4.27 | <0.001 | 1.12 |
| Mozambique | 160 | 4.98 (0.27) | 2.16 (0.15) | −2.82 | <0.001 | 1.02 |
| Rwanda | 150 | 6.78 (0.36) | 3.84 (0.24) | −2.94 | <0.001 | 0.77 |
| Zimbabwe | 164 | 5.64 (0.31) | 4.16 (0.25) | −1.49 | <0.001 | 0.41 |
| Central America | 426 | 4.40 (0.16) | 3.15 (0.13) | −1.25 | <0.001 | 0.41 |
| El Salvador | 16 | 2.63 (0.80) | 1.19 (0.42) | −1.44 | 0.17 | 0.58 |
| Guatemala | 138 | 5.54 (0.28) | 3.95 (0.28) | −1.59 | <0.001 | 0.48 |
| Honduras | 194 | 3.84 (0.25) | 3.14 (0.24) | −0.70 | 0.02 | 0.25 |
| Nicaragua | 78 | 4.15 (0.34) | 2.18 (0.28) | −1.97 | <0.001 | 0.72 |
| Asia | | | | | | |
| Philippines | 138 | 8.06 (0.34) | 5.80 (0.32) | −2.26 | <0.001 | 0.59 |
| **Positive/nurturing parenting** | | | | | | |
| Africa | 1177 | 17.63 (0.16) | 21.31 (0.15) | 3.68 | <0.001 | 0.70 |
| DRC | 163 | 17.35 (0.40) | 19.13 (0.39) | 1.78 | <0.001 | 0.35 |
| Ethiopia | 153 | 18.48 (0.44) | 22.74 (0.35) | 4.26 | <0.001 | 0.86 |
| Ghana | 192 | 16.20 (0.52) | 23.19 (0.35) | 6.99 | <0.001 | 1.14 |
| Kenya | 195 | 18.29 (0.33) | 21.74 (0.34) | 3.45 | <0.001 | 0.73 |
| Mozambique | 160 | 18.48 (0.35) | 22.52 (0.36) | 4.04 | <0.001 | 0.90 |
| Rwanda | 150 | 16.77 (0.44) | 19.36 (0.35) | 2.59 | <0.001 | 0.53 |
| Zimbabwe | 164 | 17.96 (0.40) | 20.06 (0.45) | 2.10 | <0.001 | 0.38 |
| Central America | 426 | 18.02 (0.31) | 21.50 (0.23) | 3.49 | <0.001 | 0.62 |
| El Salvador | 16 | 24.31 (1.06) | 24.19 (1.06) | −0.13 | 0.95 | 0.03 |
| Guatemala | 138 | 15.14 (0.45) | 20.18 (0.54) | 5.14 | <0.001 | 0.88 |
| Honduras | 194 | 19.45 (0.50) | 22.38 (0.20) | 2.93 | <0.001 | 0.54 |
| Nicaragua | 78 | 18.27 (0.51) | 20.96 (0.57) | 2.69 | <0.001 | 0.56 |
| Asia | | | | | | |
| Philippines | 138 | 15.99 (0.42) | 17.57 (0.42) | 1.58 | <0.001 | 0.32 |

*Note:* Effect sizes with complete cases.

Additionally, caregivers expressed high satisfaction with the program's content, group discussions and leadership of group facilitators.

Considering the compatibility of this faith-based parenting curriculum with an overwhelming majority of participating caregivers endorsing Christianity as their main faith, CF appears to be appropriate. Beyond this, parents from all regions found the attention to their faith during parent groups very helpful regardless of the caregiver's religious beliefs and reported that they would recommend the program to a friend. Overall, caregivers recognized the benefits of leveraging their faith in the best interest of their children and considered it necessary for improved parent–child relationships. Taken together, these indicators suggest the acceptability and sustainability of this program in diverse global settings.

*Feasibility*: The implementation of the CF program using a train-the-trainers model and parent program group delivery was feasible. Preliminary findings suggest that enlisting and training local leaders in the CF model promotes changes in child protection attitudes and child development knowledge through the church, faith leaders and other community influencers. This change in attitudes and norms contributes to enhanced broader outcomes across multiple levels of influence in the child's life (Bronfenbrenner, 1979). The study findings suggest that utilizing local faith and community leaders through a train-the-trainers model is sustainable and effective.

Facilitators identified both challenges and benefits of implementing the CF parent program. Most barriers, such as limitations of physical environments, and difficulties with parent engagement,

due to lack of time and family support, caregiver low education and inclement weather, were consistent with previous studies (WHO, 2022). Facilitator recommendations for improvement included providing the program to couples rather than only one caregiver, addressing psychological distress and providing economic development training to parents. Similarly, faith leaders voiced a desire for continued and expanded training beyond the delivery of CF parent groups. These concerns are corroborated by recommendations made based on a systematic review of parenting programs in LMICs (Zhang et al., 2021) which emphasized the importance of providing continued training to improve scalability, sustainability and quality assurance of parenting interventions across settings.

A unique strength of the implementation of CF is that it is built into the systems delivery supports of a large international development INGO (Britto et al., 2018). The CF model aligns with WV's local monitoring, evaluation and learning frameworks for rigorous program assessment as the parent program is situated and delivered within an established system of care that encompasses various sectors (e.g., child protection, nutrition, etc.) and mobilizes community stakeholders across ecological domains surrounding the child (Lansford et al., 2022). Within this established system of international development, and given the high endorsement of child sponsorship in participating households, future studies examining child outcomes need to examine the potential effects of these added activities on the CF overall intervention effects. The successful implementation and preliminary evaluation of this faith-based positive parenting program highlights the significant role faith-based INGOs can play in bridging community gaps and fostering environments that support flourishing in LMICs.

*Pre-post changes*: Comparison of scores pre- and post-workshop suggests that the intervention significantly improved positive parenting attitudes and reduced harsh parenting attitudes. Significant change in parent attitudes and knowledge was apparent across all countries excepting El Salvador. The extremely small sample size in El Salvador rendered the results unreliable to draw any definitive conclusions about parent program changes in this country. A significant decrease in parent-rated harsh parenting attitudes score was seen with large to medium effect sizes by country. These findings are corroborated by previous studies demonstrating that parenting programs in LMICs can be effective in improving parent attitudes and beliefs about harsh parenting and knowledge of child development, potentially reducing child maltreatment (Knerr et al., 2013; WHO, 2022; Zhang et al., 2021). Similarly, our study found statistically significant increases in positive and nurturing parenting attitudes post-parent program across all regions and countries. Nonetheless, the inclusion of more standardized measures is an area for improvement in future designs. To our knowledge, this study is among the first to examine positive parenting and the spiritual nurturance of children attitudes in caregivers from multiple countries in the Global South. Research indicates that faith-based parenting programs in LMICs can be effective when they leverage the support of religious institutions and tailor their approach to the local context (Patrick et al., 2008).

Our findings also provide preliminary evidence for not only changes in caregivers' knowledge of and attitudes toward harsh parenting and nurturing positive parenting but also changes in their use of harsh parenting practices. As predicted, pre- and 6 months after intervention *t*-test findings (improved scores from T1 to T3) in caregiver reduced use of harsh parenting practices (e.g., corporal punishment) consistently suggest that participating caregivers benefited from participating in the CF parent program. Addressing caregiver behavioral change with regards to their use of harsh

parenting practices (e.g., hitting, name-calling) is an essential precursor to promoting not only physical and emotional safety in children (WHO, 2022) but their spiritual and holistic development (CONSORTIUM, 2022). These preliminary findings align with an extensive body of research on preventive parenting interventions in the Global South aimed at reducing harsh parenting, child maltreatment and corporal punishment (Backhaus et al., 2023a). Additionally, as most evidence-based parenting programs aimed at addressing harsh parenting and maltreatment are often designed from social learning theory and cognitive-behavioral principles (Pinto et al., 2024), our findings suggests that other theoretical principles such addressing multigenerational transmission of violence, and attention to spirituality and faith in parenting, can bring about change in caregivers for whom faith is important.

In sum, the CF faith-based parent program may be feasible and appears to increase parental knowledge in relation to optimal child development and negative effects of harsh and punitive parenting. Our preliminary findings suggest the potential for shifting norms and practices underpinning violence against children and adolescents at the family level and possibly at the community level.

### Limitations

Several limitations of this preliminary study merit attention. First, we recognize that without a comparison group, relying on a pre-post design introduces bias from time trends and other potential variables influencing outcomes beyond intervention's effects. For instance, due to the nature of self-reporting, measurement of parenting knowledge and attitudes and satisfaction may be impacted by social desirability bias. These issues need to be addressed in future studies.

Notwithstanding these possibilities, our pre-post design's strength lies in the extremely short intervention period, minimizing the possibility of other time-related influences that could have led to such a significant change in outcome. Most importantly, the 6-month follow-up findings provide evidence for behavioral change (reduction in harsh parenting behavior scores), which appear to corroborate the noted change in harsh parenting attitudes and beliefs immediately after parent intervention.

Emerging research suggests that harsh parenting is more prevalent in Africa, particularly sub-Saharan Africa, than in other world regions (Devlin et al., 2018). In our study, the data seems to indicate a similar pattern, where African countries in general had the highest endorsement of harsh parenting attitudes and behaviors at baseline compared to other regions. This region also showed the highest scores changes after the CF parenting program, suggesting that CF may be most effective to implement in the African countries. However, there are some limitations to the present study which must be taken into account when interpreting these findings. For example, given the higher base rates in Africa, the larger change scores could be an effect of regression to the mean.

Second, the generalizability of the findings presented here may be limited to LMIC contexts where there is a high prevalence of Christian caregivers. Despite the relative homogeneity of caregivers' religious backgrounds, we recognize the continued need to unpack cultural differences in acceptability and parenting outcomes across the 12 countries participating in this study.

Third, considering the ambitious scaling up of the CF program, there were some unique implementation fidelity challenges. For instance, El Salvador encountered challenges in data collection and tracking of parents across time. This difficulty contributed to large caregiver attrition, and an extremely small sample resulting in

unreliable, non-significant findings. A closer examination suggests compromised research monitoring capacity rather than caregivers' rejection of the program, as exemplified by Kenya's lack of parent workshop attendance records in contrast to full data collection across all three measurement points. These challenges highlight the continued need to provide strong and intentional capacity building with regards to program evaluation, implementation and assessment (see facilitator self-reported implementation fidelity for CF parent workshops Table S7 in Supplementary Materials). Future studies should be designed to assess the implementation of CF curriculum-specific content and, ideally, implementation fidelity evaluated by independent evaluators rather than solely on the self-report of group facilitators.

Fourth, short-term parenting interventions have been demonstrated to be effective in LMIC contexts (WHO, 2022), but caregivers may also need continued support to sustain positive effects over time, perhaps in the form of booster sessions (Backhaus et al., 2023b). As mentioned previously, immediately following the workshop parents were invited to participate in 6-month-long, peer-led support groups. Preliminary reports by facilitators suggest that these groups were feasible to deliver in community settings, beneficial and well-received by parents. Additionally, the acceptability of the group-based delivery in community settings was enhanced by their ability to address caregiver-specific needs, mobilize peer support and adapt to the cultural context of the participants through support groups. Future research should carefully evaluate the implementation fidelity and efficacy of these parent-led support groups, recognizing their potential to strengthen and sustain positive changes in parent attitudes and behavior over time.

### Directions for future research

Our pre-post pilot study answers a timely call to examine whether involving local leaders in the implementation of parenting interventions can reduce violence against children. Recognizing that the effects of parenting interventions over time are mixed (Backhaus et al., 2023b), we intend to examine, in future studies, the effectiveness of this program using a casual design over time (6 months) via outcome variables including child and parent reports of behavioral change. With a rolling evaluation (multiple cohorts across 3 years) strategy designed to improve implementation quality and sustainability, we hope to have built-in feedback loops where data is collected and applied immediately to guide rapid improvements in service delivery across cohorts. We plan to test the CF program model using a randomized controlled trial in the near future.

Capacity building and awareness-raising of community and local partners promote the global Sustainable Development Goals (SDGs 5 and 16) around gender equity and just, peaceful and inclusive societies (UN-SDGs, 2024). Nonetheless, we recognize that the influence of the CF parent program can be expanded through the inclusion of couples and other primary family caretakers. In fact, in several countries in Africa, group facilitators often requested fathers be recruited and involved in the parent program. Notably, the feedback corroborates the concerns about the dangers of implicitly reinforcing gender stereotypes by assuming that mothers will be the primary recipients of parenting interventions (Morawska et al., 2021). In the future, couples/two caregiver participation in the parent program will be studied with a careful eye toward ensuring gender equity.

### Conclusion

This study leverages community partnerships with faith communities and a train-the-trainers model to implement a faith-based parenting program – *CFs* – in 12 countries in sub-Saharan Africa, Central America and South East Asia. With cautious optimism, results suggest the program was feasible to implement by local facilitators, acceptable to caregivers and effective in reducing harsh parenting attitudes and behaviors. This pilot trial adds to the evidence on holistic parenting programming to improve parenting outcomes among caregivers raising children and adolescents in the Global South. This study underscores the capacity of faith-driven parent programs to mobilize local non-specialists toward the holistic well-being of children, effectively shifting harsh parenting attitudes and beliefs and reducing harsh parenting practices to reduce child maltreatment. This study also highlights the significant role of culturally and faith-sensitive parenting practices in bridging community gaps and fostering environments that support parents and children's flourishing in LMICs.

**Open peer review.** To view the open peer review materials for this article, please visit http://doi.org/10.1017/gmh.2025.25.

**Supplementary material.** The supplementary material for this article can be found at http://doi.org/10.1017/gmh.2025.25.

**Data availability statement.** The data supporting this study's findings are available from World Vision, but restrictions apply. These data were used for the current research and are not publicly available. However, data may be available from World Vision upon reasonable request.

**Acknowledgments.** The views and opinions expressed are those of the authors and do not necessarily reflect those of World Vision. We would like to acknowledge the many churches, community centers and local leaders that supported the implementation of this parent program; our committed local project coordinators and enumerators; and the generous children, adolescents and their families who have contributed to our growing understanding of how best to support families through faith-informed parent programs.

**Author contribution.** LRF, CM, HT, RK, AM, BC, BC and TR were involved in the design of the study and protocol. AM, BC and MA are project leaders in each region who orchestrated all enumerator training, participant recruitment and data collection. RK managed online data collection and cleaning and MN and TR reviewed the manuscript and were involved in the overseeing data collection processes. PR assisted with qualitative data analyses. All authors contributed to the writing and editing of the drafts and approved the final submission.

**Financial support.** The implementation of the parent program was conducted and funded by World Vision US and World Vision International. We conducted the analysis of archival deidentified data provided by the World Vision teams; funding for analyses were provided by World Vision. However, none of the funding sources contributed to the writing of the manuscript or decision to submit it for publication. The corresponding author had full access to all the data in the study and had final responsibility for the decision to submit for publication.

**Competing interest.** The authors declare the following competing interests: LRF reports developing a Theory of Change and guidance for project implementation for World Vision, which might be perceived to create a bias toward interpreting parenting interventions as effective.

**Ethical statement.** Secondary data ethical approval was sought from the ethics committee of Fuller Theological Seminary, Pasadena, California, USA (IRB 19333854-1). Written consents were obtained from each participant prior to any data collection by World Vision.

**Trial status.** This non-randomized pilot study was not registered.

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
