## [Reviewer Report]

Overall, I congratulate the authors on this outstanding work! They have implemented and evaluated a large multi-country and multi-continent program for thousands of parents – and they have written a clean and clear manuscript to disseminate these important findings with the research community. This timely study highlights into an important area with potential for enabling broad scaling of parenting programs to many parts of the world: leveraging faith communities. I thank the authors for their excellent work.

I have several suggestions which I hope will strengthen the clarity and rigor of this paper, listed below:

Background:

• You use the terms ‘faith-driven parent programs’ and ‘faith-based parenting programs’ from the start – but you don’t clearly define these terms. Could you please provide a definition for these terms early on to ensure readers are correctly understand what you mean by ‘faith-driven parent programs’ and ‘faith-based parenting programs’? (For example, questions in my mind: Do you mean parenting programs that teach faith and religious content? Do you mean parenting programs that teach parenting content yet rely on strength from faith to learn these skills? Or do you mean standard programs led by trusted volunteers and leaders connected to a faith community?) Clarity will help readers understand your specific key point.

• There are a few claims made in the background section that could benefit from further explanation and citations:

o “There is also increasing awareness that culturally sensitive programs, particularly those that not only respect but leverage religious commitment, are needed.” – could you please add a sentence to expound on 1. What leveraging religious commitment means and 2. Why leveraging religious commitment is needed? (Perhaps as it engages a core part of people’s self-identities, community values, and/or something else? It would be highly helpful if you could also add a citation to support the increasing awareness about this – if possible.)

o “As spiritual health is an often neglected but vital aspect of child well-being” – could you please provide a citation for spiritual health being a vital aspect of child well-being? (You do provide these citations later in the introduction, but please use the citation when you initially make this claim.)

Methods:

• Data Collection: First, could you clarify when exactly T2 is? Is it immediately following the program, on the same day it ends? Or is it one day later or one week later? Second, could you clarify more about who facilitated these surveys? Was it the same person who had facilitated the program, and the same person who facilitated their baseline survey, or someone else? This is important for understanding the risk of social desirability bias, and whether participants may have felt the need to ‘report higher’ than what they initially said. Third, could you explain why this paper does not include T3 6-month follow-up data for transparency? (E.g., Is this data still being cleaned? Were follow-up rates low?)

• In your description of the intervention, could you please list the ‘Christian faith principles’ included in your intervention? You state that the intervention includes ‘Christian faith principles’ but don’t share specifically what they are; listing these principles is important to understand what is included in this intervention.

• Data analysis: You don’t specific that paired t-tests were stratified and implemented first by region of participation, and then by country of participation, but given Table 5, it seems this was the case (which is very important given differences in these locations). Could you please specify this in the methods?

• Could you do a sensitivity analysis where you consider session attendance, and model the effect on outcomes when someone attends less than 25% of sessions, then 25%-50%, 50%-74%, 75%-99%, and 100%? (You could change these categories; I propose these just as an option.) This could be an important finding since the percent who attended ALL is only 66% -- so if this program is implemented again, it is of value to know whether there is any benefit for people who attend some sessions and not all. If simpler, you could perform this analysis in a linear regression framework, as done in this pre-post paper: https://www.sciencedirect.com/science/article/pii/S2666623524000400.

Results and Discussion:

• Within the text of “Effectiveness: Caregiver Change Pre- to Post-Parent Program (T2 – T1)”, you only share the results in Africa. Please add in ALL results across all regions, since all regions are within your study, and please elaborate much more here. This is also important for people to compare where effects were largest to understand which areas this may be most effective in. From Table 5, it is clear that the effect sizes varied within countries in Africa quite meaningfully, and overall, the effect sizes were larger in African countries than Central American countries or Asian countries (especially for harsh discipline!) Please comprehensively report these results in the ‘Results’ section, and in the ‘Discussion Section’, please discuss what this nuance means for recommendations on where may be most effective to implement, rather than just stating results were statistically significant across all regions.

o A specific question to consider and hopefully unpack in the discussion: Were baseline values for harsh parenting worse at baseline in the countries with larger effect sizes? Is this why the mean differences are larger overall? Or were baseline values consistent across locations, but the actual effectiveness of the intervention truly differs between regions?

o I noted that in the discussion (under Preliminary Effectiveness), you state that results were not significant in El Salvador, with no caveat that you had an extremely small sample size in El Salvador. Please add a phrase to acknowledge the small sample which makes this result unreliable. (My reasoning: Especially for harsh parenting, your mean difference in El Salvador is consistent with other Central American countries; your p-value is just reasonably above .05 due to your small sample size. I just wish to cautious making a claim that the program in El Salvador is ineffective with this small of a sample size.)

• A major limitation is that you do not have an untreated comparison group to compare your intervention group to – which makes you reliant on a pre-post design, which is subject to bias from time trends / other potential variables which could have caused the change in outcomes, beyond the effect of your intervention. Please acknowledge this limitation in the limitations, and discuss how concerned we should be about this form of bias. A STRENGTH of your pre-post design is the intervention period is extremely short – which makes it unlikely other time trends could have taken place to lead to this sort of change in outcomes. Please discuss this, as well as any other potential reasons to reduce concern about this form of bias which is often present in pre-post studies.

• Lastly, it would be helpful for your main argument on the need for faith-based parenting programs to discuss how being faith-based changed this program, compared to if it was not faith-based? For example, here are questions in my mind (which you do not need to answer all of – but please do highlight anything noteworthy for readers): Are your effect sizes larger compared to non-faith-based parenting programming in the same region? Was recruitment faster given large faith-based networks in these majority-Christian areas? Was there better engagement because you worked within trusted communities? Were the outcomes influenced different than the outcomes that would have been influenced in non-faith based programming? Or, do you suspect this will enable further scale and reach given available networks? The authors do an excellent job demonstrating that parents found faith-based programming to be acceptable (“…benefits of leveraging their faith in the best interest of their children, and considered it necessary for improved parent-child relationships…”) -- but I wonder if the authors could go beyond assessing whether participants found it acceptable to whether it increased effectiveness.

A note for future work: With such a large sample rolled out over time, I encourage you to consider more rigorous quasi-experimental methods for further analyses in the future, such as rolled entry matching, or other methods which would enable you to compare people who were completing programming at endline (and are ‘treated’) with people who were enrolling at baseline (and are ‘untreated’). This could enable you to create a comparison group within your own sample, which would make your results much more rigorous. Additionally, if you do proceed with a pre-post design in the future, you could reduce concern about time trends by examining the baseline values over time to assess if there are meaningful changes. I do not request authors do this this prior to publication, as I suspect bias is minimal give your short intervention period – but I encourage the authors to consider these methods for future studies.

Thank you for conducting and publishing this important, creative work to strengthen families!

---

## [Reviewer Report]

This article contributes to the scarce literature on parenting programs, and especially to the literature on faith-based programs which are highly relevant in many global settings. The large number of geographic locations is also certainly a strength of the study. Authors should be commended for this large undertaking.

Abstract:

• Specify length of follow-up

• Describe results in terms of magnitude rather than only as “significant”

• Very clearly describe that preliminary effectiveness results are from treatment completers only.

Introduction: The authors provide a helpful review of the literature and articulation of the gap that this study fills. The focus on the need for faith-based programming is valuable, as this may be new to many readers. A few areas for authors to consider:

• Authors should update and expand the literature referenced on the effectiveness of parenting programs globally and in low- and middle-income countries. There are additional review papers, as well as newer randomized trials of specific programs, that would provide a fuller picture of the state of the evidence.

• Similarly, authors should update citations related to more strengths-based family interventions (beyond those focused only on parenting) that also are showing positive results. The Nurturing Families Program is one showing promise, as well as the Strengthening Families Program and READY/Tuko Pamoja in Kenya (that was also delivered in churches). These are just a few examples, but I would encourage a brief revisiting of the literature for this point and the previous point.

• I would encourage authors to be cautious in suggesting that Christianity is more amenable to promoting positive parenting than other religions. This is mainly suggested by the phrase “and Christian faith in particular,” so I would suggest removing this phrase and perhaps expanding the argument a bit to speak to how religious and spiritual traditions more broadly have potential to promote positive parenting and family relationships. I think this is a very important point but that it is important to be more inclusive in providing this rationale.

Methods:

• Minor: I suggest removing “very similar” from line 98 with having “some similarities,” as the differences are so many. The following sentences do a nice job contextualizing this.

• More detail about some methodological pieces would be helpful. Below are some questions that wouldl be helpful to answer:

• Authors describe participants were selected from those who had participated in WV activities. How were those participants found/identified (e.g., from churches?)? What types of WV activities had they participated in?

• How many were identified, and how many accepted to participate?

• Were couples, or multiple adult caregivers from the same household, encouraged to participate together? If so, how often did this happen?

• How did families identify which target child they wanted to participate if there was more than one child available?

• Why were 6-month outcomes excluded from this paper? That is a large strength of the design, so including these data would be excellent.

• How were local facilitators identified, and how did they express interest? This is important for understanding the facilitator-level outcomes. Authors should provide much more information about the faciltiators in terms of their demographics and occupations.

• Please describe how sessions were scheduled, especially given the limitation that there were logistical barriers to participation. Often church-based or community-based interventions have the advantage of sessions being held on weekends or at the convenience of participants.

• Did the fidelity checklists include the completion of the actual components of the intervention (e.g., certain didactic points made, activities facilitated), and was any of the implementation data collected via external observation, or only self-report? The current description, including in Results, sounds like this was more related to session logistics rather than content. (If this is the case, please make sure to include this in the limitations/future directions, along with the lack of ratings of quality or facilitation skills).

• It is somewhat atypical not to use standardized measures of positive and harsh parenting, including beliefs/attitudes, since there are many that have been widely used in many different contexts. Authors should perhaps mention the rationale for not using any standardized measures but, more importantly, describe how the items were developed and whether they were developed to closely match the CF curriculum.

• Qualitative data analysis: It is unclear why authors would want to avoid saturation, as that is usually the goal of analysis. The concern might have been related to having more data than needed. If so, authors should comment on how they were certain of having reached saturation without including more participant responses.

• Quantitative analysis: Please describe how any missing data were handled, state planned alpha level for determining statistical significance, and describe whether standardized effect sizes were calculated. Please clearly indicate whether analyses were based on the intent-to-treat principle and give details on whether those who did not attend as often also participated in the post-test surveys at the same rate as those with higher levels of attendance. (In Results, it is stated that all participants completed the program, so that should be specified here in the analysis plan and clearly noted as a limitation within the Discussion.)

Results

• Please provide specific fidelity results beyond that they overall followed protocols.

• For Effectiveness, it is a major limitation that the only participants were those who attended all sessions. This introduces numerous sources of bias. Please described carefully how many people enrolled in the study were excluded and the reasons why. This is very likely to inflate the effect sizes to be much larger than those reported in trials that use an intent-to-treat principle and make strong efforts to track and reach participants who attended fewer (or no) sessions of an intervention.

• For Facilitators’ Perceived Barriers, please clarify how desire for both parents to attend was a barrier.

• Given the emphasis on feasibility, authors should also include information about facilitator retention in the program, including whether there were any facilitators trained who did not go on to deliver the program. This is especially valuable to know given that the facilitators were not compensated.

Discussion

• The first paragraph of the Discussion seems to overstate the conclusions that can be made from this study. Results are encouraging, but the measures were quite light-touch (including potentially fidelity measures that were unrelated to specific content), based only on self-report, and including only participants who voluntarily completed the entire program. Authors should temper the language here and qualify their conclusions.

• The following first paragraphs of the Discussion seems to provide some new results, which need to be placed in the Results section (e.g., the qualitative quotes).

• The final sections of the Discussion present important issues, limitations, and considerations for Future Directions. Authors should revisit the feedback in this review and from other reviewers to mention further methodological limitations. Without knowing the answers to the questions in the methods above, it is unclear which aspects are limitations versus those that just require more detailed explanation.

---

## [Reviewer Report]

Thank you for the opportunity to review this revised manuscript, and many of the reviewers’ comments have been adequately addressed. I remain very encouraged by these results and impressed by the scope of this study. I look forward to seeing future research that examines efficacy and causality, and this study provides a very strong rationale for pursuing those future studies. A few concerns remain.

Most importantly, this study is set up to find large effects in many ways, as reflected in the effect sizes that are more than double what is usually reported from prevention programs. The limitations of the design and preliminary nature of these results is still not coming through clearly enough. Though the authors did add some limitations and clarifications, they are quite subtle. I have the following recommendations to frame the study in light of the design. (This is not to minimize that the results certainly do point to the promise of the intervention and rationale for future research that will test efficacy with stronger designs, such as a comparison group. Tempering the current reporting of this study will add to the credibility of those future studies.)

• Abstract

o For first statement describing the study, a more specific statement could be: As such, we conduct a pre-post study of a faith-based and community-based parenting program, Celebrating Families, in 12 countries…(to remove reliably/effectively, which can’t be concluded with the design/measures)

o I do not think you have data to support this statement since you do not have data on whether they delivered the content or their group facilitation skills (no one observed or reported on this): “We found that the trained faith- and community leaders could 11 successfully run the parent groups.”

• You could state that they could feasibly run the parent groups.

• You can also say they were successfully recruited, which is also important.

• In the Abstract, I’d simply describe that they delivered it or that you found it was feasible for leaders to deliver. In the Results/Discussion could very precisely describe what you know—feasible with successful recruitment and that they were the ones delivering when you had promising results (with the implication that they probably performed well overall, though you don’t have data on that).

• Again, this is all very promising and important preliminary data, and future studies will actually show fidelity, competency, etc.

• This is not an intent-to-treat analysis since only the treated were included.

• Report specifically that 495 people who completed baseline did not participate and were excluded from the study; this gives a better sense of the untreated group. (I believe that the 75-100% attendance rate was only limited to those who started the program, which should also be very clear.)

• The fact that these individuals are receiving other programs is important (as authors also recognize and are conducting another analysis in a separate paper to look at the potential effects of this). Please provide information about how many and what types of programs participants are participating in; this could be summarized very briefly in the text and then provided in supplemental material.

• In the discussion, I think the key limitations and the caution in interpretation should be stated up front before results are presented, with a brief but clear statement that these findings could suggest larger effects than would be found in a pre-post design. (followed quickly with a statement that results are nonetheless encouraging, providing a strong rationale for future studies that can make causal conclusions). This can be brief but clear, with the full limitations remaining where they are currently in this section. The key ones to mention up front are: pre-post design, not including participants who did not participate in the treatment, and immediate post-test.

• It is very helpful that the harsh parenting 6-month results are now included. The results for the other outcomes should also be very briefly described despite being provided in a different paper since they are so informative in terms of interpreting the implications of this study; specific numbers/results are not needed but direction of change (If there is a preprint or full publication now available, please cite.)

• Fidelity Results should be tempered by being specific about what it means that protocols were followed; readers need to be reminded that this does not mean the content of the intervention; Given lack of fidelity data, I would suggest putting this later in the results section, perhaps combined with Acceptability (optional).

• Results and Discussion sections should not include any mention of “effectiveness,” but only about pre-post changes (including the heading in Results that currently describes Effectiveness and statement in Discussion that states “and effective in reducing harsh parenting attitudes and behaviors”). Wording throughout should refer to pre-post improvements, positive pre-post changes, etc.; the additions of “suggest” and “cautious optimism” are a step in the right direction but effectiveness requires a causal design.

Minor:

• For effect size calculations, specify which standard deviation was used (pre-test or post-test)

---

## [Reviewer Report]

Thank you for your careful corrections and responses! I am happy to elect for this paper to be accepted for publication.

---

## [Editor Report]

Please correct the spelling to Sub-Saharan Africa and Southeast Asia to “sub-Saharan Africa” and “South East Asia”.